# Generating Training Data with Language Models: Towards Zero-Shot Language Understanding

**Yu Meng, Jiaxin Huang, Yu Zhang, Jiawei Han**
Department of Computer Science, University of Illinois at Urbana-Champaign
{yumeng5,jiaxinh3,yuz9,hanj}@illinois.edu

## Abstract

Pretrained language models (PLMs) have demonstrated remarkable performance in various natural language processing tasks: Unidirectional PLMs (*e.g.*, GPT) are well known for their superior text generation capabilities; bidirectional PLMs (*e.g.*, BERT) have been the prominent choice for natural language understanding (NLU) tasks. While both types of models have achieved promising few-shot learning performance, their potential for zero-shot learning has been underexplored. In this paper, we present a simple approach that uses both types of PLMs for fully zero-shot learning of NLU tasks without requiring any task-specific data: A unidirectional PLM generates class-conditioned texts guided by prompts, which are used as the training data for fine-tuning a bidirectional PLM. With quality training data selected based on the generation probability and regularization techniques (label smoothing and temporal ensembling) applied to the fine-tuning stage for better generalization and stability, our approach demonstrates strong performance across seven classification tasks of the GLUE benchmark (*e.g.*, 72.3/73.8 on MNLI-m/mm and 92.8 on SST-2), significantly outperforming zero-shot prompting methods and achieving even comparable results to strong few-shot approaches using 32 training samples per class[1].

## 1 Introduction

Pretrained language models (PLMs) [5, 8, 11, 19, 34, 40, 41] have achieved human-level performance on natural language understanding (NLU) tasks [66, 67] when fine-tuned on a large amount of task-specific training data. However, such a supervised fine-tuning paradigm is drastically different from how humans perform these tasks: We barely need to see many task-specific training samples to perform well. Recently, many studies have revealed the intriguing few-shot learning potential of PLMs: By converting task descriptions to natural language prompts and injecting them into PLMs, prompt-based approaches [5, 13, 55, 56, 59] leverage task-specific information for better training data efficiency and have achieved remarkable few-shot results.

When prompt-based methods are applied to the zero-shot setting, however, the PLMs' predictions are much less accurate. For example, GPT-3's zero-shot performance is much degraded relative to its few-shot performance [5], especially on challenging tasks like natural language inference (NLI). Without any task-specific samples, it is indeed challenging for PLMs to effectively interpret the prompts that come in different formats and are unseen in the pretraining data. To familiarize PLMs with various prompts for zero-shot generalization to unseen tasks, a recent study proposes instruction tuning [70], which fine-tunes PLMs on a large collection of different tasks described by instructions. Despite its strong performance, its success is grounded in the large number of cross-task annotated datasets (*e.g.*, train on many non-NLI tasks and transfer to NLI tasks) and the gigantic model size (*e.g.*, hundreds of billions of parameters), posing great challenges for training and using them.

---

[1]Code can be found at `https://github.com/yumeng5/SuperGen`.

In this work, we study zero-shot learning of PLMs on NLU tasks without any task-specific or cross-task data. Motivated by the strong text generation power of recent PLMs [5, 23, 30, 52], we propose SuperGen, a **Super**vision **Gen**eration approach, wherein training data are created via a unidirectional PLM (*i.e.*, the generator) which generates class-conditioned texts guided by label-descriptive prompts. A bidirectional PLM (*i.e.*, the classifier) is then fine-tuned on the generated texts to perform the corresponding task. Both PLMs can be of moderate size to fit in typical research hardware (*e.g.*, a GPT-2-sized [51] generator and a RoBERTa$_{Large}$-sized [34] classifier). With supervision automatically created by the generator, SuperGen eliminates the need for task-specific annotations and provides the classifier PLM with a larger amount of training data than in few-shot scenarios. We call such a setting zero-shot because the entire process does not need any human annotated data, either from the target task or other tasks. The major difference from previous methods is that we synthesize training data for the target task, whereas existing zeros-shot methods do not use any form of training data from the test domain (but may train on other domains) and directly perform inference on the target task.

Across seven classification tasks of the GLUE benchmark [66], SuperGen significantly outperforms the prompt-based zero-shot method and even achieves an overall better result in both average performance and stability than strong few-shot approaches that use 32 annotated samples per class. We identify several key factors to the strong performance of SuperGen through ablation studies: (1) selecting quality training data based on their generated probability, and (2) using label smoothing and temporal ensembling to regularize fine-tuning on generated data.

## 2 Related Work

### 2.1 Few-Shot and Zero-Shot Learning with PLMs

Instead of using a large amount of annotated training data for fine-tuning PLMs on downstream tasks, few-shot learning studies how to better leverage only a small amount of task-specific training data, a more realistic scenario in many applications. The most strict few-shot learning setting does not assume access to any unlabeled data or large validation sets for hyperparameter tuning [48], where prompt-based methods [5, 13, 33, 35, 55–57, 59, 63, 84] are prominently deployed to inject task descriptions into PLMs and make effective use of their language modeling capability for improved training data efficiency in low-data regimes. More broadly, semi-supervised learning additionally leverages unlabeled task-specific data, where data augmentation [7, 73], regularization [43] and bootstrapping [56] methods are commonly used.

Zero-shot learning, on the other hand, is a much more challenging setting with absolutely no access to any task-specific data. When prompt-based methods are directly used to obtain predictions from PLMs without any training, their zero-shot performance can be much worse [5, 13]—difficult NLU tasks can be barely formulated as prompts that resemble the format of pretraining data, posing great challenges for PLMs to accurately interpret and leverage the prompts without given any training samples. The current mainstream of zero-shot learning is based on transfer learning: By converting a set of tasks with abundant annotations into instruction templates [42, 54, 70, 74], entailment pairs [79, 80] or question-answer formats [50, 86] and fine-tuning PLMs on them, the PLMs acquire the cross-task transfer ability [78] to execute unseen tasks when they are formulated in a similar format. Our work proposes a different approach from these studies: We use a unidirectional PLM to generate training data for fine-tuning another PLM on the target task. This not only removes the need for a large amount of cross-task annotations, but also eliminates the task difference in training and inference. Moreover, different from previous studies [1, 76] that rely on labeled data to fine-tune the generative PLM, we directly use prompts to guide data generation without fine-tuning.

### 2.2 Controlled Text Generation with PLMs

Controlled text generation [22] aims to steer the generated texts of language models towards desired contents, styles or domains. Through fine-tuning PLMs on attribute-specific data, high-level control (*e.g.*, generating certain topics or sentiments [88]), fine-grained control (*e.g.*, generating specific words or phrases [6]) or both [24] can be achieved. Adapting PLMs to generate texts of specific attributes can also be realized at inference time without any further training of the PLMs [10, 26, 27, 32, 47, 75]. Different text attributes can also be represented during pretraining time as control codes [23] which later can serve as explicit guidance for generating domain/attribute-specific texts.

The idea of generating category-conditioned texts as training data has been explored for topic classification with bag-of-words or LSTM-based language models [38, 39], which may not have

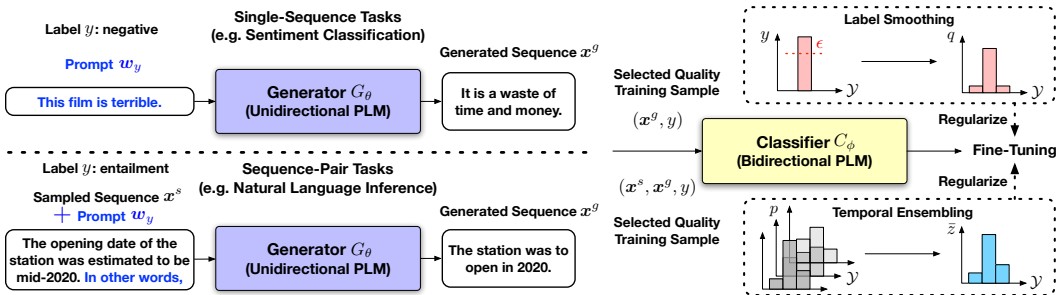

Figure 1: Overview of SuperGen for zero-shot learning of NLU tasks. A unidirectional PLM generates training data guided by label-descriptive prompts. Quality training samples are selected based on average log generation probability. A bidirectional PLM is fine-tuned on the selected training set with label smoothing and temporal ensembling as regularization to perform the classification task.

enough capacity to generate quality training data for challenging NLU tasks. With more powerful PLMs, the idea of using prompts as guidance has emerged recently: Since natural language generation is largely based on contexts, using certain prompts to start a sequence can effectively steer the subsequent texts to be generated. The prompts can be either in natural language [57] or as learnable parameters [31]. In this work, we also guide text generation via prompts, but for the novel purpose of creating training data for NLU tasks. There have been studies with similar goals, such as generating similar/dissimilar sentences for training sentence embeddings [58] and using labeled samples as demonstrations to prompt large PLMs [81] for creating novel training data. In this work, we explore generating training data *without using any labeled samples* for a wide range of different NLU tasks. The similar setting is also explored in a concurrent study [77]. Compared to annotated task-specific data, the generated texts may contain noise and have domain difference from the downstream task. We introduce several important strategies for effective fine-tuning on generated data.

# 3    Method

## 3.1    Preliminaries

**Problem Formulation.**    We consider solving a classification problem[2] where we are only given the label space $\mathcal{Y}$ and a mapping $\mathcal{M} : \mathcal{Y} \to \mathcal{W}$ that converts each label $y \in \mathcal{Y}$ into a label-descriptive prompt (*i.e.*, a short phrase) $\boldsymbol{w}_y \in \mathcal{W}$. We assume access to a unidirectional PLM $G_\theta$ as the generator and a bidirectional PLM $C_\phi$ which will be fine-tuned as the classifier[3]. We also assume the pretraining corpus $\mathcal{D}$ (*e.g.*, Wikipedia) is available. Fig. 1 shows an overview of our proposed SuperGen method.

**Text Generation with Unidirectional PLMs.**    A unidirectional PLM $G_\theta$ is pretrained to maximize the generation probability of each token in a sequence $\boldsymbol{x} = [x_1, x_2, \ldots, x_n]$ conditioned on previous tokens:

$$\max_\theta \prod_{i=1}^n p_\theta(x_i|\boldsymbol{x}_{<i}), \quad \text{where} \quad p_\theta(x_i|\boldsymbol{x}_{<i}) = \frac{\exp(\boldsymbol{e}_i^\top \boldsymbol{h}_i)}{\sum_{j=1}^{|V|} \exp(\boldsymbol{e}_j^\top \boldsymbol{h}_i)}.$$

Here, $p_\theta(\cdot)$ is usually parameterized using token embeddings $\boldsymbol{e}$ and contextualized embeddings $\boldsymbol{h}$ given by a Transformer [65] encoder.

After pretraining, $G_\theta$ can be directly used to generate new texts by recursively sampling tokens from its output probability distribution. Typically, a temperature hyperparameter $\tau > 0$ is introduced during sampling [20] to adjust the sharpness of the probability distribution:

$$p_\theta(x_i|\boldsymbol{x}_{<i}) = \frac{\exp(\boldsymbol{e}_i^\top \boldsymbol{h}_i/\tau)}{\sum_{j=1}^{|V|} \exp(\boldsymbol{e}_j^\top \boldsymbol{h}_i/\tau)}, \tag{1}$$

---

[2]We do not consider regression tasks in this work due to the difficulty of generating texts conditioned on a continuous label space. However, there exist approaches [14, 53] that solve regression tasks by training on classification tasks. We leave the integration of SuperGen with these methods as future work for solving regression tasks.

[3]We assume the classifier to be bidirectional PLMs since they generally work better than unidirectional PLMs in NLU tasks; we can in principle use any PLM as the classifier.

where $\tau \to 0$ approximates greedily picking the most probable next token; $\tau \to \infty$ induces a uniform distribution. Additionally, sampled tokens can be confined to the top-$k$ most probable ones to avoid low-quality tokens. In this work, we find such top-$k$ sampling with temperature is sufficient to produce coherent and meaningful texts as training data for NLU tasks. Exploring more sophisticated sampling strategies [21] is left for future work.

## 3.2 Training Data Generation

When given a label-descriptive prompt such as "Write a negative review:", humans are able to produce texts pertaining to the corresponding class. We aim to leverage the strong text generation power of a unidirectional PLM $G_\theta$ for the same purpose of creating class-conditioned training data. We note that $G_\theta$ is directly used for generation without any parameter updates. The prompts used for different NLU tasks in GLUE are summarized in Table 1.

**Generating Single Sequences.** For single-sequence NLU tasks such as sentiment classification (*e.g.*, SST-2), we simply use a prompt $\boldsymbol{w}_y$ corresponding to label $y$ as the beginning of the sequence and let $G_\theta$ generate the remaining sequence:

$$\boldsymbol{x}^g \leftarrow G_\theta(\boldsymbol{w}_y),$$

where $G_\theta(\boldsymbol{w}_y)$ denotes using $\boldsymbol{w}_y$ as the input to $G_\theta$ and recursively sampling tokens from the distribution in Eq. (1) until a full sequence is generated; $\boldsymbol{x}^g$ denotes the *generated* sequence (*i.e.*, excluding the prompt), which will be paired with $y$ to form one training sample $(\boldsymbol{x}^g, y)$.

Table 1: Prompts used to generate class-conditioned texts for different GLUE tasks. SST-2 is a single-sequence classification task and the rest are sequence-pair classification tasks. Generation for CoLA does not use prompts but by varying sampling temperatures. $\boldsymbol{x}^s$ denotes a sequence randomly sampled from the pretraining corpus; $\boldsymbol{x}^g$ denotes the sequence to be generated by $G_\theta$; ... denotes skipping at least one sequence. See Appendix A for more details.

| Task | Label | Prompt |
|------|-------|--------|
| **SST-2** | positive | Rating: 5.0 $\boldsymbol{x}^g$ |
| | negative | Rating: 1.0 $\boldsymbol{x}^g$ |
| **MNLI** | entailment | $\boldsymbol{x}^s$. In other words, $\boldsymbol{x}^g$ |
| | neutral | $\boldsymbol{x}^s$. Furthermore, $\boldsymbol{x}^g$ |
| | contradiction | There is a rumor that $\boldsymbol{x}^s$. However, the truth is: $\boldsymbol{x}^g$ |
| **QNLI** | entailment | $\boldsymbol{x}^s$? $\boldsymbol{x}^g$ |
| | not entailment | $\boldsymbol{x}^s$? ... $\boldsymbol{x}^g$ |
| **RTE** | entailment | $\boldsymbol{x}^s$. In other words, $\boldsymbol{x}^g$ |
| | not entailment | $\boldsymbol{x}^s$. Furthermore, $\boldsymbol{x}^g$ |
| **MRPC** | equivalent | $\boldsymbol{x}^s$. In other words, $\boldsymbol{x}^g$ |
| | not equivalent | $\boldsymbol{x}^s$. Furthermore, $\boldsymbol{x}^g$ |
| **QQP** | equivalent | $\boldsymbol{x}^s$? In other words, $\boldsymbol{x}^g$ |
| | not equivalent | $\boldsymbol{x}^s$? Furthermore, $\boldsymbol{x}^g$ |

For syntactic tasks like linguistic acceptability classification (*e.g.*, CoLA) which requires generating both linguistically acceptable and unacceptable sequences, we start the sequence with random stop words and use varying sampling temperatures for generating different sequences. A smaller temperature (*e.g.*, $\tau = 0.1$ in Equation (1)) sharpens the sampling probability distribution towards the most probable tokens, thus the resulting sequence is more likely to be linguistically acceptable. Using a larger temperature (*e.g.*, $\tau = 10$ in Equation (1)) flattens the sampling probability distribution to be more uniform, and the generated tokens will be nearly random, which can create linguistically incorrect sequences.

**Generating Sequence Pairs.** Sequence-pair classification tasks require generating two sequences of specific relationships (*e.g.*, entailment, contradiction). We sample[4] the first sequence $\boldsymbol{x}^s$ from the pretraining corpus $\mathcal{D}$, concatenate the prompt $\boldsymbol{w}_y$ with $\boldsymbol{x}^s$, and generate the second sequence $\boldsymbol{x}^g$:

$$\boldsymbol{x}^g \leftarrow G_\theta\left([\boldsymbol{x}^s; \boldsymbol{w}_y]\right), \ \boldsymbol{x}^s \sim \mathcal{D}.$$

The sequence pair training sample will then be formed as $(\boldsymbol{x}^s, \boldsymbol{x}^g, y)$.

**Rewarding and Penalizing Repetitions for Sequence Pair Generation.** A common issue in text generation is degenerate repetition [21, 23, 51, 71] where generated texts get stuck in repetition loops. To address this issue, one approach is to discourage repetition by reducing the logits of tokens that are already in the sequence before performing sampling [23]. In sequence pair generation, however, it is sometimes desirable to encourage the second sequence to repeat some words in the first sentence (*e.g.*, for generating an entailment or a paraphrase). Therefore, we propose a simple modification of

---

[4]In principle, we can also generate the first sequence using $G_\theta$, but we find sampling from $\mathcal{D}$ improves the diversity of texts.

Eq. (1) that rewards/penalizes repetition based on whether the token has appeared in $\boldsymbol{x}^s/\boldsymbol{x}^g$:

$$p_\theta(x_i|\boldsymbol{x}_{<i}) = \frac{\exp(\boldsymbol{e}_i^\top \boldsymbol{h}_i/\omega)}{\sum_{j=1}^{|V|} \exp(\boldsymbol{e}_j^\top \boldsymbol{h}_i/\omega)}, \quad \text{where} \quad \omega = \begin{cases} \tau\alpha & x_i \in \boldsymbol{x}^s \wedge x_i \notin \boldsymbol{x}^g \\ \tau\beta & x_i \in \boldsymbol{x}^g \\ \tau & \text{else} \end{cases}, \quad (2)$$

and $\alpha > 0, \beta > 0$ are hyperparameters. By setting $\alpha < 1$ and $\beta > 1$, we can promote tokens in $\boldsymbol{x}^s$ that have not appeared in $\boldsymbol{x}^g$ to have a higher chance of being generated, and discourage the generation of repetitive tokens in $\boldsymbol{x}^g$ to mitigate degenerate repetition. The parameters used for different tasks are listed in Appendix B Table 9.

## 3.3 Effective Fine-Tuning on Generated Texts

With the generated training data, one can fine-tune a bidirectional PLM $C_\phi$ as the classifier to perform the NLU task. However, training $C_\phi$ via standard supervised training on all generated texts is likely to yield suboptimal performance on downstream tasks because (1) the generated texts may contain noise as $G_\theta$ may not always produce texts pertaining to the desired class, especially for challenging sequence pair tasks with subtle semantic relationships; and (2) the generated texts can be considered as originated from the domain of $G_\theta$'s pretraining data, with a potentially different distribution from the downstream task; straightforward application of supervised training will result in overfitting to the pretraining domain and diminishing generalization ability, a common challenge in transfer learning [64, 87]. To address these challenges, we next introduce several simple and important strategies for more effective and stable fine-tuning on generated texts.

**Selecting Quality Training Data.** We aim to select generated texts $\boldsymbol{x}^g$ that are most likely to pertain to the desired label $y$ (*i.e.*, with the highest $p(\boldsymbol{x}^g|y)$). The true probability $p(\boldsymbol{x}^g|y)$ is unknown and we estimate it via the generation probability given by $G_\theta$ conditioned on the prompt $\boldsymbol{w}_y$:

$$p(\boldsymbol{x}^g|y) \approx p_\theta(\boldsymbol{x}^g|\boldsymbol{w}_y) = \prod_{i=1}^{n} p_\theta\left(x_i\big|[\boldsymbol{w}_y; \boldsymbol{x}_{<i}^g]\right).$$

Since the above measure is biased towards shorter sequences, we instead use the geometric mean of the above conditional generation probability (or equivalently, the average log probability) of all tokens in $\boldsymbol{x}^g$ as the ranking score, following [82]:

$$r = \frac{1}{n} \sum_{i=1}^{n} \log p_\theta\left(x_i\big|[\boldsymbol{w}_y; \boldsymbol{x}_{<i}^g]\right). \quad (3)$$

To construct a training set consisting of $N$ samples per class, we will generate more samples (*e.g.*, $10N$), and select training data based on the score $r$ in Eq. (3): For all tasks except CoLA, the top-$N$ ones of each class are selected; for CoLA, the top-$N$ ones are used as linguistically acceptable training samples, and the bottom-$N$ ones as linguistically unacceptable sequences.

**Regularization for Better Generalization and Stability.** Even with the above training data selection procedure, the resulting training set may still contain noise and there exists domain difference from the downstream tasks. We apply two regularization techniques, *label smoothing* [62] and *temporal ensembling* [28] for better fine-tuning stability and generalization.

Given a training sample $(\boldsymbol{x}^g, y)$, *label smoothing* trains the classifier $C_\phi$ to minimize the standard cross-entropy loss between the label and the classifier's prediction $p_\phi(\boldsymbol{x}^g)$, except that the label is a weighted average of the one-hot vector and a uniform distribution over all labels:

$$\min_\phi -\sum_{j=1}^{|\mathcal{Y}|} q_j \log(p_\phi(\boldsymbol{x}^g)_j), \quad (4)$$

where $q_j = \mathbb{1}(j = y)(1 - \epsilon) + \epsilon/|\mathcal{Y}|$ and $\epsilon$ is the smoothing weight. By forcing the classifier to be less confident on training data, label smoothing improves robustness to label noise [36] and prevents overfitting to the training set [44], thus improving generalization to different domains.

The motivation for *temporal ensembling* is that neural networks usually first pick up easy and general patterns in the data before learning more sophisticated and dataset-specific features [83], and thus the

earlier states of the network offer better generalizability to different domains. We therefore record the predictions $\boldsymbol{p}_\phi = p_\phi(\boldsymbol{x}^g)$ of $C_\phi$ on each training sample $(\boldsymbol{x}^g, y)$ at different training steps, and use the accumulated moving-average predictions $\bar{\boldsymbol{z}}$ to regularize the latest model training. This also helps suppress the fluctuation in model predictions due to data noise, offering better noise-robustness [45]. We update ensembled predictions $\bar{\boldsymbol{z}}$ once every $B$ batches:

$$\hat{\boldsymbol{z}} \leftarrow \gamma\hat{\boldsymbol{z}} + (1 - \gamma)\boldsymbol{p}_\phi, \ \bar{\boldsymbol{z}} \leftarrow \hat{\boldsymbol{z}}/(1 - \gamma^t), \tag{5}$$

where $\hat{\boldsymbol{z}}$ has a zero initialization; $\gamma$ is the momentum parameter; $t$ is the number of updates $\bar{\boldsymbol{z}}$ has received; the division $(1 - \gamma^t)$ is for bias correction [28]. We also use the ensembled prediction $\bar{\boldsymbol{z}}$ as a reliable signal to filter out noisy training samples: Only those samples on which $\bar{\boldsymbol{z}}$ strongly agrees with the label $y$ (*i.e.*, $\bar{z}_y > \delta$ where $\delta > 0$ is a threshold parameter) will be used for training.

We regularize model training by extending Eq. (4) to add a KL divergence regularization term from the model prediction to the ensembled prediction weighed by $\lambda$:

$$\min_\phi -\sum_{j=1}^{|\mathcal{Y}|} q_j \log(p_\phi(\boldsymbol{x}^g)_j) - \lambda \sum_{j=1}^{|\mathcal{Y}|} \bar{z}_j \log \frac{p_\phi(\boldsymbol{x}^g)_j}{\bar{z}_j}. \tag{6}$$

We follow [28] to slowly ramp-up $\lambda$ during training.

### 3.4 Overall Algorithm

We summarize SuperGen for single-sequence NLU tasks in Algorithm 1. Solving sequence-pair problems follows the same algorithm except the pretraining corpus $\mathcal{D}$ is needed for sampling the first sequence $\boldsymbol{x}^s$.

## 4 Experimental Setup

**Downstream Tasks and Metrics.** We use all the tasks included in GLUE [66] except STS-B which is a regression task. Please refer to Appendix C for more details about GLUE tasks. We follow the evaluation protocol of [13]: We use F1 score as the metric for QQP and MRPC, Matthews correlation for CoLA, and accuracy for the rest of the tasks. The original development sets of these tasks are used for testing. For all reported results, we include the average and standard deviation over 5 different random seeds.

**Models.** Unless specified otherwise, we use CTRL (1.63B parameters) [23] as the generator $G_\theta$ and COCO-LM$_{\text{Large}}$ (367M parameters) [40] as the classifier $C_\phi$. We also show the results using similar-sized PLMs (GPT-2 [51]/RoBERTa [34]) as the generator/classifier in Section 5.6.

**Fine-Tuning Settings and Hyperparameters.** We note that SuperGen is compatible with any fine-tuning method; while using more sophisticated methods may grant

---

**Algorithm 1:** SuperGen for Zero-Shot Learning.

**Input:** $\mathcal{Y}$: Label space; $\mathcal{P}$: Label-descriptive prompts; $G_\theta$: Unidirectional PLM; $C_\phi$: Bidirectional PLM.

**Parameter:** $N$: Number of training samples per class to generate; $M(\gg N)$: Number of total training samples to generate; $T$: Number of training steps; $B$: Ensemble prediction update interval; $\delta$: Threshold parameter.

**Output:** $C_\phi^*$: Classifier that classifies input texts into $\mathcal{Y}$.

for $y \in \mathcal{Y}$ do
  $\mathcal{T}_y \leftarrow \{\}$
  // Class $y$ train set init.
  for $i \in [1, 2, \ldots, M]$ do
    $\boldsymbol{x}^g \leftarrow G_\theta(\boldsymbol{w}_y)$
    $\mathcal{T}_y \leftarrow \mathcal{T}_y \bigcup \{(\boldsymbol{x}^g, y)\}$
  end
end
$\mathcal{T} \leftarrow \{\}$
// Selected train set.
for $y \in \mathcal{Y}$ do
  Sort $\mathcal{T}_y$ in descending order by Eq. (3)
  $\mathcal{T} \leftarrow \mathcal{T} \bigcup \mathcal{T}_y[: N]$
end
$\hat{\boldsymbol{z}} \leftarrow \boldsymbol{0}$
// Ensembled prediction init.
$\mathcal{T}^* \leftarrow \mathcal{T}$
// Filtered train set.
for $i \in [1, 2, \ldots, T]$ do
  Fine-tune $C_\phi$ via Eq. (6) on a minibatch of $\mathcal{T}^*$
  if $i\%B = 0$ then
    Update $\hat{\boldsymbol{z}}, \bar{\boldsymbol{z}}$ via Eq. (5)
    $\mathcal{T}^* \leftarrow \{(\boldsymbol{x}^g, y)|\bar{z}_y > \delta, (\boldsymbol{x}^g, y) \in \mathcal{T}\}$
  end
end
**return** $C_\phi^* = C_\phi$

---

ible with any fine-tuning method; while using more sophisticated methods may grant further performance improvement, we use the basic prompt-based fine-tuning with manual templates approach for simplicity and clarity. For all tasks, we use the same templates and label words as in [13]. Under the zero-shot learning setting, it is not possible to tune hyperparameters due to the lack

Table 2: Results on seven GLUE classification tasks. We report average and standard deviation (as subscripts) performance over 5 different random seeds. $\dagger$: Results from LM-BFF [13].

| Method | MNLI-(m/mm) (Acc.) | QQP (F1) | QNLI (Acc.) | SST-2 (Acc.) | CoLA (Matt.) | RTE (Acc.) | MRPC (F1) | AVG |
|---|---|---|---|---|---|---|---|---|
| **Zero-Shot Setting:** No task-specific data (neither labeled nor unlabeled). | | | | | | | | |
| Prompting$^\dagger$ | $50.8_{0.0}/51.7_{0.0}$ | $49.7_{0.0}$ | $50.8_{0.0}$ | $83.6_{0.0}$ | $2.0_{0.0}$ | $51.3_{0.0}$ | $61.9_{0.0}$ | 50.1 |
| SuperGen | $\mathbf{72.3}_{0.5}/\mathbf{73.8}_{0.5}$ | $\mathbf{66.1}_{1.1}$ | $\mathbf{73.3}_{1.9}$ | $\mathbf{92.8}_{0.6}$ | $\mathbf{32.7}_{5.5}$ | $65.3_{1.2}$ | $82.2_{0.5}$ | **69.4** |
| $-$ data selection | $63.7_{1.5}/64.2_{1.6}$ | $62.3_{2.2}$ | $63.9_{3.2}$ | $91.3_{2.0}$ | $30.5_{8.8}$ | $62.4_{1.5}$ | $81.6_{0.2}$ | 65.1 |
| $-$ label smooth | $70.7_{0.8}/72.1_{0.7}$ | $65.1_{0.9}$ | $71.4_{2.5}$ | $91.0_{0.9}$ | $9.5_{1.0}$ | $64.8_{1.1}$ | $\mathbf{83.0}_{0.7}$ | 65.2 |
| $-$ temporal ensemble | $62.0_{4.6}/63.6_{4.8}$ | $63.9_{0.3}$ | $72.4_{2.0}$ | $92.5_{0.9}$ | $23.5_{7.0}$ | $63.5_{1.0}$ | $78.8_{2.2}$ | 65.3 |
| **Few-Shot Setting:** Use 32 labeled samples/class (half for training and half for development). | | | | | | | | |
| Fine-tuning$^\dagger$ | $45.8_{6.4}/47.8_{6.8}$ | $60.7_{4.3}$ | $60.2_{6.5}$ | $81.4_{3.8}$ | $\mathbf{33.9}_{14.3}$ | $54.4_{3.9}$ | $76.6_{2.5}$ | 59.1 |
| Manual prompt$^\dagger$ | $68.3_{2.3}/70.5_{1.9}$ | $65.5_{5.3}$ | $64.5_{4.2}$ | $92.7_{0.9}$ | $9.3_{7.3}$ | $69.1_{3.6}$ | $74.5_{5.3}$ | 63.6 |
| $+$ demonstration$^\dagger$ | $\mathbf{70.7}_{1.3}/\mathbf{72.0}_{1.2}$ | $\mathbf{69.8}_{1.8}$ | $69.2_{1.9}$ | $92.6_{0.5}$ | $18.7_{8.8}$ | $68.7_{2.3}$ | $77.8_{2.0}$ | 66.9 |
| Auto prompt$^\dagger$ | $68.3_{2.5}/70.1_{2.6}$ | $67.0_{3.0}$ | $68.3_{7.4}$ | $92.3_{1.0}$ | $14.0_{14.1}$ | $\mathbf{73.9}_{2.2}$ | $76.2_{2.3}$ | 65.8 |
| $+$ demonstration$^\dagger$ | $70.0_{3.6}/72.0_{3.1}$ | $67.7_{5.8}$ | $68.5_{5.4}$ | $\mathbf{93.0}_{0.6}$ | $21.8_{15.9}$ | $71.1_{5.3}$ | $\mathbf{78.1}_{3.4}$ | **67.3** |
| Fully supervised$^\dagger$ | *89.8/89.5* | *81.7* | *93.3* | *95.0* | *62.6* | *80.9* | *91.4* | *84.9* |

Table 3: Results with different groups of prompts. CoLA does not use prompts for generation. The number of prompt groups is equal to the number of the task labels.

| Prompt Group | MNLI-(m/mm) | QQP | QNLI | SST-2 | RTE | MRPC |
|---|---|---|---|---|---|---|
| #0 (Original) | $\mathbf{72.3}_{0.5}/\mathbf{73.8}_{0.5}$ | $66.1_{1.1}$ | $\mathbf{73.3}_{1.9}$ | $\mathbf{92.8}_{0.6}$ | $65.3_{1.2}$ | $\mathbf{82.2}_{0.5}$ |
| #1 | $70.7_{1.4}/72.4_{1.2}$ | $65.5_{1.4}$ | $71.9_{1.7}$ | $92.2_{0.9}$ | $64.4_{1.6}$ | $81.9_{0.4}$ |
| #2 | $70.8_{0.6}/72.1_{0.8}$ | $65.6_{1.1}$ | $72.2_{2.2}$ | $92.4_{0.8}$ | $64.7_{1.8}$ | $81.8_{0.8}$ |
| #3 | $70.9_{1.4}/72.2_{1.4}$ | - | - | - | - | - |
| Mixed | $72.2_{0.7}/73.4_{0.6}$ | $\mathbf{66.9}_{1.5}$ | $73.0_{1.7}$ | $\mathbf{92.8}_{0.9}$ | $\mathbf{66.3}_{1.0}$ | $81.3_{2.0}$ |

of validation sets. Therefore, we keep all fine-tuning hyperparameters (*e.g.*, learning rate, batch size, training epochs, number of generated training samples, label smoothing and temporal ensembling hyperparameters) the same across all tasks. See Appendix B Table 10 for details.

**Compared Methods and Ablations.** We include the results of zero-shot prompting, standard few-shot fine-tuning and the four few-shot prompt-based fine-tuning methods proposed in [13]. We also conduct ablation studies by removing the following three techniques from SuperGen one at a time: (1) not using Eq. (3) for training data selection but randomly selecting the same amount of training data ($-$ data selection); (2) not using label smoothing ($-$ label smooth) but using one-hot labels; and (3) not using temporal ensembling (*i.e.*, using Eq. (4) instead of Eq. (6) as the training objective) ($-$ temporal ensemble). Lastly, we include the fully supervised fine-tuning results trained on the entire training sets.

## 5 Evaluation

### 5.1 Main Results

We present the results of SuperGen, its ablations and compared methods in Table 2. Overall, SuperGen significantly outperforms zero-shot prompting and achieves an overall better result than all few-shot methods. Notably, SuperGen results in much smaller variance over different random seeds than few-shot approaches on most tasks—with access to more training data, fine-tuning of PLMs becomes much more stable. The ablation results demonstrate that all three strategies (*i.e.*, quality training data selection, label smoothing and temporal ensembling) play important roles in improving and stabilizing the final performance, especially on challenging tasks like MNLI.

### 5.2 Using Different Prompts

One important factor of SuperGen is the choice of label-descriptive prompts as they directly influence the quality of generated training samples. To study the impact of different prompt choices on the

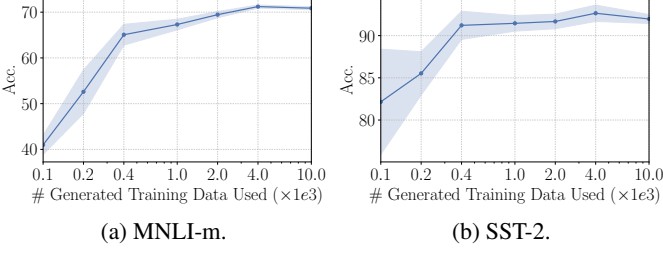

(a) MNLI-m.  (b) SST-2.

Figure 2: Classifier accuracy fine-tuned on different amount of generated training data (after data selection). Dots and error bars are the average performance and the standard deviation over 5 seeds, respectively.

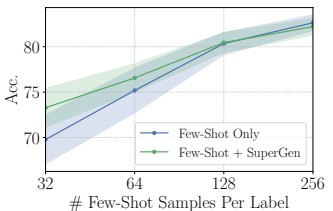

Figure 3: Classifier accuracy on MNLI-m fine-tuned on the few-shot samples only vs. on the few-shot and SuperGen generated set with varying few-shot set sizes.

final model performance, we create different groups of prompts other than the original ones. We replace the prompt for one label used in Table 1 with a synonymous one and keep other prompts unchanged when forming a different prompt group (Please refer to Appendix A Table 8 for details). We also experiment with mixing the generated data by different prompt groups (mixed). The results are shown in Table 3. Overall, the model performance under different prompts is quite close, except on RTE whose test set is very small, potentially resulting in the higher variance. In this work, we manually choose simple prompts that make intuitive sense, and we leave the automatic searching of optimal prompts as future work.

## 5.3 Results with Different Amount of Generated Data

With training data automatically created by the generator, we can have a virtually infinite amount of training samples. We show the results of using different amount of generated data (after quality data selection) for fine-tuning the classifier $C_\phi$ in Fig. 2 on MNLI-m and SST-2. When the number of training data is small (*e.g.*, $100$), the fine-tuning variance is high, resulting in the similar instability issue with few-shot settings. With more generated data used, both average performance and training stability improve, yielding comparable results (with smaller variance) to fine-tuning using few-shot task-specific data. However, when too many generated data (*e.g.*, $10,000$) are used, the classifier's performance slightly drops, probably due to increased label noise—recall that the training data are selected based on the ranking score in Eq. (3), so using more data results in the inclusion of more lower-ranking texts in the training set and reduced data quality. One way to address this issue is to use a fixed selection ratio and increase the total number of generated texts to obtain a larger number of high-quality training data. However, this comes at a greater computation cost in the generation step. An important future direction is thus to develop better data selection strategies.

## 5.4 Using SuperGen in Few-Shot Settings

We present a simple extension of SuperGen to few-shot settings and show that the generated data of SuperGen may also improve the few-shot performance. When few-shot samples are available, we first fine-tune the classifier on the few-shot training set (standard prompt-based fine-tuning without regularization), and then continue fine-tuning the classifier on the generated data by SuperGen as described in Section 3.3. This allows the classifier to effectively leverage the knowledge from the few-shot training set to filter out noisy samples in the generated data, as temporal ensembling regularizes the classifier to remember the predictions learned previously and only keeps samples on which the model predictions agree with the label. We show the benefits of incorporating generated data for different few-shot sample sizes on MNLI in Fig. 3 (we use half labeled samples for classifier training and half for development): When the few-shot training and validation sets are rather small ($32 - 64$ samples per label in total), fine-tuning the classifier on the SuperGen generated set further (after fine-tuning on the few-shot samples) brings notable performance improvements. However, such benefits diminish with more few-shot training samples: The generated data fail to improve the few-shot performance when there are $128$ samples per label, and even worsen the classifier performance with $256$ samples per label. This is probably because our synthetic data generation process is zero-shot and does not leverage any few-shot samples; the resulting generated samples may not be of high enough quality to boost the few-shot performance when there are relatively abundant annotated samples. Possible ways to use few-shot samples for generation include using them as demonstrations [5], for creating augmentations [29] and for tuning the generators. We leave the explorations of generating higher quality data by leveraging few-shot samples for future work.

Table 4: Comparisons with using CTRL for zero-shot prompting and for knowledge distillation. [†]: The entire training set is used as unlabeled data.

| Method | MNLI-(m/mm) | SST-2 |
|---|---|---|
| SuperGen | $\mathbf{72.3}_{0.5}/\mathbf{73.8}_{0.5}$ | $\mathbf{92.8}_{0.6}$ |
| CTRL Prompting | $38.5_{0.0}/39.2_{0.0}$ | $72.5_{0.0}$ |
| Knowledge Distill[†] | $40.8_{0.5}/41.5_{0.6}$ | $73.6_{0.8}$ |

Table 5: Results with different generator/classifier PLMs.

| PLMs ($G_\theta/C_\phi$) | MNLI-(m/mm) | SST-2 |
|---|---|---|
| CTRL/COCO-LM | $\mathbf{72.3}_{0.5}/\mathbf{73.8}_{0.5}$ | $92.8_{0.6}$ |
| CTRL/RoBERTa | $69.0_{0.8}/70.6_{0.9}$ | $\mathbf{93.0}_{1.5}$ |
| GPT-2/COCO-LM | $69.5_{1.2}/71.3_{1.3}$ | $88.2_{1.8}$ |
| GPT-2/RoBERTa | $68.3_{0.9}/69.7_{0.7}$ | $88.6_{0.8}$ |

## 5.5 Using Generators for Knowledge Distillation

Apart from using unidirectional PLMs $G_\theta$ for training data generation, one could also directly apply them to unlabeled data formulated as prompts to obtain zero-shot predictions (*i.e.*, prompting [5, 13]), which can then be used as soft labels to train the classifier $C_\phi$. In Table 4, we show (1) the zero-shot prediction accuracy of CTRL (the best out of three different prompts, details in Appendix D) and (2) the classifier performance trained from CTRL's predictions on the entire unlabeled training set as soft labels (*i.e.*, knowledge distillation). Similar to the observations in previous studies [5, 70, 85], the zero-shot predictions of unidirectional PLMs are quite inaccurate and directly using them as soft labels to train classifiers does not yield good results. We hypothesize that the advantages of using unidirectional PLMs for training data generation over using them for zero-shot predictions are twofold: (1) Better flexibility in prompt formats. When unidirectional PLMs are used for zero-shot predictions, the prompts have to be designed so that the label word is the last token in the sequence to be predicted, as unidirectional PLMs cannot attend to subsequent tokens. Such constraints may result in the prompt being dissimilar to the pretraining data distribution and worsen the prediction quality of the PLMs. On the contrary, using unidirectional PLMs for generation is not subject to any prompt format constraints. (2) More direct uses of PLMs' language modeling ability. Using unidirectional PLMs for training data generation *directly* leverages the PLMs' output token probability. Applying PLMs for zero-shot prediction, however, requires an additional step to convert token predictions to label predictions (*i.e.*, the verbalizer [56]), and such a mapping process usually necessitates manual curation and can hardly be optimal [13] especially without abundant task-specific data.

## 5.6 Using Different PLMs

The final performance of SuperGen is relevant to the choice of PLMs as the generator/classifier. Apart from the default PLM choice, we report the results of using GPT-2$_{XLarge}$ (1.54B parameters) [51] as the generator and RoBERTa$_{Large}$ (356M parameters) [34] as the classifier in Table 5 with everything else unchanged. When using GPT-2, we change the prompt used for SST-2 to "The film is bad/terrible/awful." for the negative label and "The film is good/great/excellent." for the positive label, since the original prompts used for SST-2 in Table 1 are a part of the control codes of CTRL and cannot be effectively leveraged by GPT-2. Overall, both CTRL and GPT-2 are able to generate quality training data for good fine-tuned classifier performance; CTRL consistently yields better results than GPT-2 regardless of the choice of the classifier PLM, probably because CTRL is pretrained with control codes which provide explicit guidance for generating texts of certain domains and attributes. We also observe that the generated text quality is strongly correlated to the generator's model size—using a smaller version of GPT-2 (*e.g.*, with 117M parameters) results in significantly less coherent texts and can hardly serve as training data. An interesting future direction is to try larger generator PLMs (*e.g.*, GPT-3) which may create training data of better quality.

## 5.7 Case Studies

We present concrete examples of generated texts guided by prompts of different labels in Table 6. The generated sequences are not only coherent, but also pertain to the corresponding labels. For easier tasks like SST-2, the generated texts almost always correctly reflect the desired sentiment polarity specified by the prompt. For more difficult tasks like MNLI, sometimes the generated texts are not of the correct label (Appendix E Table 13 shows some negative results). The existence of such label noise motivates our use of the regularization techniques in the fine-tuning stage. In the future, it will be interesting to develop new methods to better control text generation towards the desired label.

Table 6: Example generated texts for SST-2, MNLI and QQP. *Sampled sequences* from pretraining corpus ($x^s$) are italicized; generated sequences ($x^g$) are underlined; **prompts** ($w^y$) are in bold.

| Task | Label | Generated Text |
|------|-------|----------------|
| SST-2 | positive | **Rating: 5.0** The film is a great example of the kind of movie that you can watch over and over. |
|  | negative | **Rating: 1.0** The film was a total waste of time. I would not recommend this movie to anyone. |
| MNLI | entailment | *The construction of the station began in 2016, when the opening date was estimated to mid-2020.* **In other words**, the station was to open in 2020. |
|  | neutral | *The construction of the station began in 2016, when the opening date was estimated to mid-2020.* **Furthermore**, it is expected that a new bus terminal will be built at this station. |
|  | contradiction | **There is a rumor that** *The construction of the station began in 2016, when the opening date was estimated to mid-2020.* **However, the truth is:** The construction started in 2017, and the official opening date was set for March 31, 2018. |
| QQP | equivalent | *What are the most wear resistant steels?* **In other words,** what are the most durable steels? |
|  | not equivalent | *What are the most wear resistant steels?* **Furthermore,** what is the best way to clean them? |

# 6 Discussions and Conclusions

**Ethical Considerations.** While PLMs have demonstrated remarkable text generation and understanding capability, they can come with potential risks or harms [2, 3, 5] such as generating misinformation [46] or amplifying harmful biases [49]. The focus of our work is on utilizing existing PLMs to generate training data for NLU tasks instead of developing new PLMs or generation methods. Therefore, our method can be used in company with any bias reduction and correction techniques [15, 37] to mitigate the risks of PLMs.

**Limitations.** One inherent limitation with zero-shot learning is the lack of access to task-specific samples for hyperparameter tuning, whereas the performance of neural networks is usually heavily dependent on the choice of hyperparameters even when the training algorithm and training set are fixed [48]. Also, without access to any labeled data, the generated training data quality may not be high enough to achieve good performance on challenging tasks, especially when the task distribution is significantly different from the pretraining data distribution (*e.g.*, the "linguistically incorrect" label of CoLA requires generating sequences with grammar mistakes – a different distribution from the one used to train PLMs). A promising direction to address the above limitations is extending SuperGen to few-shot settings (*e.g.*, the setting studied in Section 5.4) and leveraging a small amount of labeled data for generating better quality data and for hyperparameter tuning.

**Conclusions.** We propose SuperGen, an automatic supervision generation approach for zero-shot learning of NLU tasks. By providing label-descriptive prompts as guidance to a unidirectional PLM, training data can be automatically created for fine-tuning a bidirectional PLM. Our framework differs from previous transfer-learning-based zero-shot methods in that SuperGen does not rely on cross-task annotations and eliminates the task difference in training and inference. We show that several strategies are important for effective and stable fine-tuning on generated data, including quality training data selection, label smoothing and temporal ensembling. SuperGen achieves strong performance on seven classification tasks of the GLUE benchmark, even yielding comparable or better results than sophisticated few-shot learning methods and offering better stability. There is large room for future work, including but not limited to: Extension to few-shot learning settings, exploring larger generator models [25, 68], better fine-tuning techniques to leverage generated data and better strategies for selecting quality training data.

# Acknowledgments

Research was supported in part by US DARPA KAIROS Program No. FA8750-19-2-1004 and INCAS Program No. HR001121C0165, National Science Foundation IIS-19-56151, IIS-17-41317, and IIS 17-04532, and the Molecule Maker Lab Institute: An AI Research Institutes program supported by NSF under Award No. 2019897, and the Institute for Geospatial Understanding through an Integrative Discovery Environment (I-GUIDE) by NSF under Award No. 2118329. Any opinions, findings, and conclusions or recommendations expressed herein are those of the authors and do not necessarily represent the views, either expressed or implied, of DARPA or the U.S. Government. Yu Meng is supported by the Google PhD Fellowship. We thank anonymous reviewers for valuable and insightful feedback.

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
