# A  Details of Prompts Used for Different Tasks

Table 7: Extensions of Table 1 with more details of prompts used to generate class-conditioned texts for different GLUE tasks. SST-2 and CoLA are single-sequence classification tasks and the rest are sequence-pair classification tasks. Generation for CoLA does not use prompts but by varying sampling temperatures. Text generation with CTRL [23] requires starting with control codes, and we use the ones that correspond to the pretraining corpus where the first sequence is sampled: For MNLI, RTE and MRPC, the first sequence is sampled from Wikipedia; for QNLI and QQP, the first sequence is sampled from OpenWebText [17]. $x^s$ denotes a sequence randomly sampled from the pretraining corpus; $x^g$ denotes the sequence to be generated by $G_\theta$; ... denotes skipping at least one sequence. The prompts used for SST-2 are part of the CTRL [23] codes.

| Task | Task Type | Control Code | Label | Prompt |
|---|---|---|---|---|
| **SST-2** | single-sequence | Reviews | positive
negative | Rating: 5.0 $x^g$
Rating: 1.0 $x^g$ |
| **CoLA** | single-sequence | Links | grammatical
not grammatical | $x^g$
$x^g$ |
| **MNLI** | sequence-pair | Wikipedia | entailment
neutral
contradiction | $x^s$. In other words, $x^g$
$x^s$. Furthermore, $x^g$
There is a rumor that $x^s$. However, the truth is: $x^g$ |
| **QNLI** | sequence-pair | Links | entailment
not entailment | $x^s$? $x^g$
$x^s$? ... $x^g$ |
| **RTE** | sequence-pair | Wikipedia | entailment
not entailment | $x^s$. In other words, $x^g$
$x^s$. Furthermore, $x^g$ |
| **MRPC** | sequence-pair | Wikipedia | equivalent
not equivalent | $x^s$. In other words, $x^g$
$x^s$. Furthermore, $x^g$ |
| **QQP** | sequence-pair | Links | equivalent
not equivalent | $x^s$? In other words, $x^g$
$x^s$? Furthermore, $x^g$ |

Table 8: Different prompt groups used in the experiments of Section 5.2. We replace the original prompt for each label with an alternative one and keep other prompts unchanged when forming a different prompt group.

| Task | Label | Original | Alternative |
|---|---|---|---|
| **SST-2** | positive
negative | Rating: 5.0 $x^g$
Rating: 1.0 $x^g$ | Rating: 4.0 $x^g$
Rating: 2.0 $x^g$ |
| **MNLI** | entailment
neutral
contradiction | $x^s$. In other words, $x^g$
$x^s$. Furthermore, $x^g$
There is a rumor that $x^s$. However, the truth is: $x^g$ | $x^s$. To put it another way, $x^g$
$x^s$. In addition, $x^g$
People believe that $x^s$. However, the truth is: $x^g$ |
| **QNLI** | entailment
not entailment | $x^s$? $x^g$
$x^s$? ... $x^g$ | Question: $x^s$? Answer: $x^g$
Question: $x^s$? Answer: ... $x^g$ |
| **RTE** | entailment
not entailment | $x^s$. In other words, $x^g$
$x^s$. Furthermore, $x^g$ | $x^s$. To put it another way, $x^g$
$x^s$. In addition, $x^g$ |
| **MRPC** | equivalent
not equivalent | $x^s$. In other words, $x^g$
$x^s$. Furthermore, $x^g$ | $x^s$. To put it another way, $x^g$
$x^s$. In addition, $x^g$ |
| **QQP** | equivalent
not equivalent | $x^s$? In other words, $x^g$
$x^s$? Furthermore, $x^g$ | $x^s$? To put it another way, $x^g$
$x^s$? In addition, $x^g$ |

We present more details about the prompts used for different tasks in Table 7 which is an extended version of Table 1.

For SST-2, we fix the beginning of the generated sequence $x^g$ to be "The/this film/movie" to make sure the generated texts are related to movie reviews. For CoLA, we start the generated sequence $x^g$ with a random stop word. For QNLI and QQP, the first sequence is always a question, and we require the sampled sequence $x^s$ to end with a question mark and begin with one of the following words: "how", "what", "why", "who", "which", "where", "when", "whom", "whose". For QNLI, the generated sequence $x^g$ for the "entailment" label is the one that immediately follows the sampled sequence $x^s$; the generated sequence $x^g$ for the "not entailment" label is randomly sampled from the paragraph following $x^g$ excluding the first sequence that immediately follows $x^g$.

Table 9: Hyperparameters for generating training data of different tasks. $\tau$: Temperature during sampling ($\tau = 0$ means using greedy sampling); $\alpha$ and $\beta$: Repetition rewarding/penalizing parameters; $M$: Number of total generated texts per label. The top-$k$ sampling (if $\tau > 0$) uses $k = 10$.

| Task | Label | $\tau$ | $\alpha$ | $\beta$ | $M$ |
|------|-------|--------|----------|---------|-----|
| **SST-2** | positive | 0.2 | - | 1.2 | 25,000 |
| | negative | | - | 1.2 | 25,000 |
| **CoLA** | grammatical | [0.1, 10] | - | 1.2 | 10,000 |
| | not grammatical | | - | 1.2 | 10,000 |
| **MNLI** | entailment | 0 | 0.8 | 1.1 | 25,000 |
| | neutral | | 1.3 | 1.3 | 25,000 |
| | contradiction | | 1.1 | 1.1 | 25,000 |
| **QNLI** | entailment | 0 | 0.9 | 1.2 | 25,000 |
| | not entailment | | 0.9 | 1.2 | 25,000 |
| **RTE** | entailment | 0 | 0.8 | 1.1 | 30,000 |
| | not entailment | | 1.1 | 1.1 | 30,000 |
| **MRPC** | equivalent | 0 | 0.8 | 1.1 | 30,000 |
| | not equivalent | | 1.1 | 1.1 | 30,000 |
| **QQP** | equivalent | 0 | 1.0 | 1.2 | 25,000 |
| | not equivalent | | 1.2 | 1.2 | 25,000 |

Table 10: Hyperparameters used for fine-tuning on different tasks (they are kept same for all tasks). Fine-tuning-related hyperparameters (*e.g.*, learning rate, batch size) follow the default values (when the validation set is not available) in Appendix A of [13]; regularization-related hyperparameters follow the default values in label smoothing and temporal ensembling. $lr$: Learning rate; $bs$: Batch size; $N|\mathcal{Y}|$: Total number of selected generated data (*i.e.*, training set size); $B$: Ensemble prediction update interval; $T$: Number of training steps; $\epsilon$: Label smoothing parameter; $\gamma$: Temporal ensembling momentum parameter; $\delta$: Threshold for filtering out noisy data; $\lambda_{\text{max}}$: Maximum weight (after ramp-up) of temporal ensembling regularization.

| $lr$ | $bs$ | $N|\mathcal{Y}|$ | $B$ | $T$ | $\epsilon$ | $\gamma$ | $\delta$ | $\lambda_{\text{max}}$ |
|------|------|-------|-----|-----|------------|----------|----------|------------------------|
| 1e-5 | 16 | 6,000 | 100 | 1,125 | 0.15 | 0.8 | 0.8 | 10 |

We also show the different prompt groups used in the experiments of Section 5.2 in Table 8.

## B   Hyperparameters and Reproducibility

**Hyperparameters for Generating Training Data.**   Table 9 lists the hyperparameters used in the training data generation stage. For sequence-pair tasks, we use greedy sampling for better reproducibility. For labels that require generating entailment, paraphrase, or equivalent sequence pairs, we set $\alpha \leq 1$ to encourage word overlapping between the second sequence and the first sequence; otherwise, we set $\alpha = \beta > 1$ to discourage word repetition.

To construct a training set consisting of $N$ samples per class, we will generate $M$ samples per class, and select training data based on the score $r$ in Eq. (3): For all tasks except CoLA and the "neutral" label of MNLI, the top-$N$ ones of each class are selected; for CoLA, the top-$N$ ones are used as the training sample as linguistically acceptable sequences, and the bottom-$N$ ones are as linguistically unacceptable sequences; for the "neutral" label of MNLI, we find it better to randomly select $N$ samples from the total $M$ samples instead of using the ranking score, probably because a neutral hypothesis with respect to the premise has a wide range of possibilities (*i.e.*, any hypothesis that is not entailed by or contradicts with the premise will be neutral), and random selection improves the diversity in generated hypotheses of the neutral label.

**Hyperparameters for Fine-Tuning.**   Table 10 lists the hyperparameters used in the fine-tuning stage. We keep them the same across all tasks except CoLA which uses $\delta = 0$ because half of the training data for CoLA are intentionally made to be of low quality (*i.e.*, as linguistically unacceptable sequences) and there is no need to filter them out. We follow [28] to slowly ramp-up $\lambda$ in Equation (6)

Table 11: Different prompts used on MNLI for CTRL zero-shot prompting and knowledge distillation baselines. $x_1$ and $x_2$ denote the first and second input sequence, respectively.

| Prompt | Template | Label name |
|---|---|---|
| #1 | Sentence 1: $x_1$ Sentence 2: $x_2$
Does Sentence 1 entail Sentence 2?
The answer is: | entailment: Yes
neutral: Maybe
contradiction: No |
| #2 | Premise: $x_1$ Hypothesis: $x_2$
Does the premise entail the hypothesis?
Options: Yes. No. Maybe. The answer is: | entailment: Yes
neutral: Maybe
contradiction: No |
| #3 | Premise: $x_1$ Hypothesis: $x_2$
What is the relation between the premise and the hypothesis?
Options: Entailment. Neutral. Contradiction. The answer is: | entailment: Entailment
neutral: Neutral
contradiction: Contradiction |

Table 12: Different prompts used on SST-2 for CTRL zero-shot prompting and knowledge distillation baselines. $x$ denotes the input sequence.

| Prompt | Template | Label name |
|---|---|---|
| #1 | $x$ This is | positive: good; negative: bad |
| #2 | $x$ It was | positive: good; negative: bad |
| #3 | Review: $x$ Sentiment: | positive: Positive; negative: Negative |

during the first 10 ensembles: $\lambda(t) = \lambda_{\max} \exp(-5(1 - t/10)^2)$ where $t$ is the number of prediction ensembles performed.

**Computation Environment.** All experiments are conducted on NVIDIA GeForce RTX 3090 GPUs. SuperGen can be run on typical research hardware (*e.g.*, with $> 10\text{GB}$ GPU memory). The generator PLM $G_\theta$ does not need to be trained so a relatively large generator can be used (*e.g.*, a 1.63B-parameter CTRL model).

## C    GLUE Tasks

We provide the details of the seven classification tasks included in the GLUE benchmark.

**MNLI:** Multi-genre Natural Language Inference [72] aims to predict whether a given premise sentence entails, contradicts or neutral with respect to a given hypothesis sentence. It has two test sets, matched (MNLI-m) and mismatched (MNLI-mm), which correspond to samples from the same sources as the training set and samples that do not resemble the training data, respectively.

**QQP:** Quora Question Pairs [60] aims to determine whether a pair of questions asked are semantically equivalent.

**QNLI:** Question Natural Language Inference aims to predict whether a given sentence contains the answer to a given question sentence.

**SST-2:** Stanford Sentiment Treebank [61] aims to determine if a movie review has positive or negative sentiment.

**CoLA:** Corpus of Linguistic Acceptability [69] aims to determine whether a given sentence is linguistically acceptable or not.

**RTE:** Recognizing Textual Entailment [4, 9, 16, 18] aims to predict whether a given premise sentence entails a given hypothesis sentence or not.

**MRPC:** Microsoft Research Paraphrase Corpus [12] aims to predict whether two sentences are semantically equivalent or not.

## D  Knowledge Distillation Baseline Details

We show the concrete prompts used for the knowledge distillation baseline in Tables 11 and 12 on MNLI and SST-2, respectively. We use the best prompt (prompt #1 in both tables) out of the three according to the zero-shot test set prediction accuracy for generating soft labels to train the classification model (*i.e.*, knowledge distillation). The classifier is trained with Kullback–Leibler (KL) divergence as the objective to approximate the soft labels generated by CTRL on the entire training set.

## E  Negative Generation Results

Table 13: Negative example generated texts for MNLI that do not pertain to the desired label. *Sampled sequences* from pretraining corpus ($x^s$) are italicized; generated sequences ($x^g$) are underlined; **prompts** ($w^y$) are in bold. In example #1, the generated text corresponding to the "entailment" label actually pertains to the "neutral" label. In example #2, the generated text corresponding to the "contradiction" label actually pertains to the "neutral" label. In example #3, the generated text corresponding to the "neutral" label actually pertains to the "contradition" label.

| Example | Label | Generated Text |
|---|---|---|
| #1 | entailment | *Directed by Daniel J. Sullivan, the cast features Kelly AuCoin, Annie Parisse, Lee Tergesen and Alex Wolff.* **In other words**, the cast features a lot of people who are not in the original Broadway cast. |
| | neutral | *Directed by Daniel J. Sullivan, the cast features Kelly AuCoin, Annie Parisse, Lee Tergesen and Alex Wolff.* **Furthermore**, it was announced that a sequel to "The Last Airbender" will be released in 2019. |
| | contradiction | **There is a rumor that** *Directed by Daniel J. Sullivan, the cast features Kelly AuCoin, Annie Parisse, Lee Tergesen and Alex Wolff.* **However, the truth is:** The cast includes only two actors from the original Broadway production. |
| #2 | entailment | *As a result, minor problems begin to look serious and rare events begin to seem common.* **In other words**, a minor problem can begin to look serious and rare events seem common. |
| | neutral | *As a result, minor problems begin to look serious and rare events begin to seem common.* **Furthermore**, the number of people who are affected by these changes is increasing. |
| | contradiction | **There is a rumor that** *As a result, minor problems begin to look serious and rare events begin to seem common.* **However, the truth is:** The only thing that can cause such an effect is a major change in the weather. |
| #3 | entailment | *The character was announced on 20 September 2019, but casting wasn't revealed until 10 December 2019.* **In other words**, the casting wasn't revealed until 10 December 2019. |
| | neutral | *The character was announced on 20 September 2019, but casting wasn't revealed until 10 December 2019.* **Furthermore**, it is unknown whether the role will be played by a male or female actor. |
| | contradiction | **There is a rumor that** *The character was announced on 20 September 2019, but casting wasn't revealed until 10 December 2019.* **However, the truth is:** The character was announced on 19 October 2018. |

Table 13 demonstrates concrete examples where the generated texts may not correctly pertain to the label of the prompt. The existence of such label noise motivates our use of the regularization techniques in the fine-tuning stage. We believe that larger generator PLMs (*e.g.*, GPT-3 [5]) can bring about better text generation quality and improve the accuracy in producing texts that pertain to the desired class. Furthermore, better filtering strategies can be developed in the future to select training data with the correct labels.