# OpenReview forum: "Generating Training Data with Language Models: Towards Zero-Shot Language Understanding"
_NeurIPS.cc/2022/Conference — NeurIPS 2022 Accept_

### Official Review · Reviewer_V4Sv · 2022-07-08

**Rating:** 6
**Confidence:** 3
**Soundness:** 3 good
**Presentation:** 3 good
**Contribution:** 3 good

**Summary:**

They study zero-shot learning of PLMs on NLU tasks without any task-specific or cross-task data.

They propose SuperGen approach: training data are created via unidirectional PLM. A bidirectional PLM is then finetuned on the generated texts to perform tasks.

Across seven classification tasks of the GLUE benchmark, SuperGen significantly outperforms the prompt-based zero-shot method. Their method for selecting quality training data based on their generated probability and using label smoothing and temporal ensembling demonstrates the strong performance of SuperGen.

**Questions:**

- In Table2, the performance of SuperGen without label smoothing for CoLA drops a lot. Is there a specific reason for that?
- Generator selection: Why did you not use large size PLMs (e.g., GPT3) from the beginning for the high-quality generation, if the model size matters for high-quality text generation like you mentioned in page 8, line 280?
- Are there other regularization methods other than label smoothing and temporal ensembling that you might have considered?

**Strengths And Weaknesses:**

Strengths

- Supervision automatically created by the generator eliminates the need for task-specific annotations.
- Rewarding and penalizing repetitions for sequence pair generation reduces the chance that the generated texts get stuck in repetition loops.
- Reasonable method for selecting quality training data to take the average log probability.
- Label smoothing forces the classifier to have less probability on training data. That improves robustness to label noise and prevent overfitting to the training set.

Weaknesses
- Since prompt would be a key factor for the performance, it would be nice to have prompt ablation studies for each task, not just three tasks(MNLI, RTE, MRPC).
- Even though SuperGen significantly outperforms zero-shot prompting and achieves a better result than all the few-shot methods, the performance cannot match to their SOTA results.
- The generated texts are not judged by humans.

---

> ### Author Response · Authors · 2022-08-02
> **Response to Reviewer V4Sv**
>
> Thanks for your thoughtful review and feedback! As mentioned in the general response, we have updated the paper with the prompt ablation studies on all tasks, as you suggest.
> We provide more detailed discussions as follows:
> 1. **The performance of SuperGen does not match the SOTA few-shot results.** In this work, we focus on zero-shot learning without using any task-specific data. Our comparisons to few-shot methods are not intended to be apple-to-apple (the few-shot methods have access to labeled task data whereas SuperGen does not), but to demonstrate the possibility of generating high-quality training data in a zero-shot manner to achieve strong results on NLU tasks. When SuperGen has access to few-shot labeled data, a simple method that trains classifiers first on few-shot samples and then on our generated data further improves the performance, as shown by the newly added results in Section 5.5 (suggested by Reviewer kf6s). We believe that training data generation under the few-shot setting will be an interesting and important future direction to improve the few-shot SOTA. We have added some discussions regarding this at the end of Section 5.5.
> 2. **The generated texts are not judged by humans.** We did not conduct human evaluations for the generated data because it is challenging to manually evaluate the quality of the synthetic data – many different aspects can impact the classifier performance trained on the generated data, such as their accuracy, diversity and closeness to the target task distribution – some of these aspects are hard to be quantitatively rated by humans. Therefore, we believe that the final classifier performance is still the most direct and reliable metric to evaluate the quality of the generated data. We agree that it is important to develop systematic human evaluation metrics for the synthetic data, and we would like to leave this as future work.
> 3. **The performance of SuperGen without label smoothing for CoLA drops a lot. Is there a specific reason for that?** One possible reason is that CoLA contains a “linguistically incorrect” class (i.e., sentences with grammar mistakes), which corresponds to a rather different data distribution from the pretraining data distribution that PLMs are trained on. Therefore, it is challenging to use PLMs for generating linguistically incorrect sentences in a zero-shot manner – the synthesized “linguistically incorrect” sentences will not be so similar to those in the CoLA test set. Without label smoothing, the classifier may overfit to the specific patterns in the synthesized data and perform poorly on the real linguistically incorrect sentences from the test set.
> 4. **Why not use larger size PLMs (e.g., GPT3)?** We believe that using larger PLMs indeed will produce better quality generated data. However, since large PLMs like GPT3 are usually not freely/publicly available and cannot be easily used in typical research hardware, we choose moderate-sized PLMs (can fit in GPUs with >10GB memory) in our experiments for reproducibility.
> 5. **Are there other regularization methods other than label smoothing and temporal ensembling that you might have considered?** We have also tried using noise-robust loss functions such as generalized cross entropy (Zhang et al.) and mean absolute error (Ghosh et al.), but using those loss functions does not yield better results than using label smoothing and temporal ensembling, so we end up not changing the loss function but applying regularization.
>
> Thank you again for your review! Please let us know if you have any further comments.
>
> References:
> Zhang, Zhilu et al. “Generalized Cross Entropy Loss for Training Deep Neural Networks with Noisy Labels.” NeurIPS 2018.
> Ghosh, Aritra et al. “Robust Loss Functions under Label Noise for Deep Neural Networks.” AAAI 2017.

---

> > ### Comment · Reviewer_V4Sv · 2022-08-09
> > **comment**
> >
> > Thank you for your reply. I have read other reviewers' thoughts. I would like to keep my score to 6.

---

> > > ### Author Response · Authors · 2022-08-09
> > > **Thanks for your feedback!**
> > >
> > > Thanks for your feedback!

---

### Official Review · Reviewer_kf6s · 2022-07-10

**Rating:** 6
**Confidence:** 4
**Soundness:** 3 good
**Presentation:** 3 good
**Contribution:** 4 excellent

**Summary:**

This paper proposes generating training data for certain NLU tasks using a pre-trained generative model and using this data to fine-tune a pre-trained MLM for the target task. The generated data is filtered based on likelihood and the classifier is regularized using label smoothing for robustness to noise. They generate on the order of a few thousand examples using prompts and the label space for 7 GLUE tasks and compare against zero-shot and few-shot baselines.

**Questions:**

Have you considered generating synthetic data using the few-shot examples? Do you think that would make it possible to generate higher quality data with a greater value-add?
E.g. https://arxiv.org/pdf/2102.01335.pdf

**Limitations:**

I think the limitations section doesn't really discuss potential societal impact. Perhaps the synthetic data generated by the model might contain unwanted biases that could be harmful.

**Strengths And Weaknesses:**

Strengths:
The proposed approach to generate data using prompts is quite interesting and is generating data of good enough quality to achieve non-trivially good performance on the GLUE tasks. This has important implications for showing that using LLMs, zero- and few-shot settings don’t need to be as such by augmenting them with synthetic data, at least for the kinds of tasks explored in this paper.

Weaknesses:
Calling this entire setup “zero-shot” is a bit of a stretch. The generator can be seen as zero-shot for generating task-specific examples, but the fine-tuned classifier not so much. For this reason, the truly zero-shot baseline seems weak and it is unsurprising that the classifier fine-tuned on synthetic data outperforms it. Perhaps an interesting comparison that was left out was fine-tuning on few-shot data vs. fine-tuning on a combination of few-shot and the synthetic data to quantify how much value is added by the synthetic data.

There’s nothing new and surprising about using synthetic data, it’s more the approach to generating the synthetic data that is the interesting contribution here. The story (and title) could use some revision to clarify this.

---

> ### Author Response · Authors · 2022-08-02
> **Response to Reviewer kf6s**
>
> Thanks for your thoughtful review and feedback! As mentioned in the general response, we have updated the paper with the few-shot results by fine-tuning the classifier on a combination of few-shot samples and synthesized data, as you suggest.
> We provide more detailed discussions as follows:
> 1. **Regarding the wording “zero-shot”.** We agree that the definition of zero-shot is open to different interpretations. In this work, we mainly follow the definition in Wei et al. which calls fine-tuning a language model “zero-shot” as long as the training process does not use any task-specific data (it could use data from other domains/distributions). We have clarified this in the abstract and introduction. We would greatly appreciate any further suggestions to make this clearer if necessary.
> 2. **Regarding the added few-shot results.** In Section 5.5, we present a simple method to leverage both few-shot samples and synthesized ones for classifier fine-tuning. The promising results verify the benefits of the synthesized training data. We believe that the results can be further improved if the few-shot samples are also leveraged in the data generation stage.
> 3. **Generate higher quality data using few-shot samples?** Thanks for pointing out the reference; we have added it to the updated version. We agree that this is a great direction to improve the generation quality. Specifically, the few-shot samples may be used as demonstrations, for creating augmentations and for tuning the generators. We would like to leave the explorations along this line as future studies.
> 4. **The synthetic data generated by the model might contain unwanted biases.** We agree that this is a valid concern, and we have discussed this in the “Ethical Considerations” section instead of the “Limitations” section. Please let us know if you have further concerns regarding this.
>
> Thank you again for your review! Please let us know if you have any further comments.
>
> Reference:
> Wei, Jason et al. “Finetuned Language Models Are Zero-Shot Learners.” ICLR 2022.

---

> > ### Comment · Reviewer_kf6s · 2022-08-09
> > **Re: Author Response**
> >
> > Thanks for including the few shot results -- I do think they help to make the paper better. It might also be interesting to show a trend of what happens as you gradually increase the number of real examples i.e. at what point does the synthetic data stop contributing and/or start hurting performance. Please note, this is not something you need to do right away: just a suggestion for the next version or camera ready if the paper is accepted.
> >
> > I disagree that the FLAN definition of zero-shot applies here. In their case, they take great care to ensure that the eval sets are very different tasks from the ones used for instruction-tuning (see Section 2.2 in their paper). As far as I understand, in this work the goal is to generate synthetic data for the task that is being evaluated. So the settings in the two papers are different. It is worth emphasizing that there's no issue with the setup -- the fine-tuned classifier is just not zero-shot.

---

> > > ### Author Response · Authors · 2022-08-09
> > > **Response to Reviewer kf6s**
> > >
> > > Dear Reviewer kf6s,
> > >
> > > Thank you very much for the suggestions! We will include the results of the model performance vs. number of real examples in the next version of the paper.
> > >
> > > Regarding the zero-shot definition, we have updated the paper with more clarifications. Specifically, we have added the following explanation to the Introduction section:
> > > > We note that the generator creates synthetic samples in a zero-shot manner, and the classifier is fine-tuned on the synthetic data (the classifier is thus not zero-shot, but there are no task-specific data required in such a process).
> > >
> > > We hope this helps clarify our setup further. Please let us know if you have any further suggestions!
> > >
> > > Thank you again for your review and feedback!

---

> > > > ### Comment · Reviewer_kf6s · 2022-08-09
> > > > **Discussion on zero-shot framing**
> > > >
> > > > The statement in the update doesn't recognize that the synthetic data is the task-specific data for the classifier. The whole paper will probably need a few rounds of revision for reframing the entire story and shifting it away from calling it zero-shot (starting with updating the title). This is not a suggestion to try to revise the paper before the rebuttal period is over! Take some time to think it over carefully and consider incorporating these suggestions for the next major revision/camera-ready if it is accepted.

---

### Official Review · Reviewer_XfNA · 2022-07-17

**Rating:** 5
**Confidence:** 4
**Soundness:** 2 fair
**Presentation:** 3 good
**Contribution:** 3 good

**Summary:**

This paper proposes SuperGen, which targeting at automatic dataset generation for zero-shot learning of NLU tasks. The framework relies on manually designed prompts to "extract" task-specific knowledge from a pre-trained PLM, thus constructing a synthetic dataset. In the whole process, no human annotations are required. The experiments show that the proposed method can even outperform few-shot learning methods on some NLU tasks.

**Questions:**

"we include the average and standard deviation over 5 different random seeds."
Are these 5 runs performed on 5 different few-shot train/dev sets or on the same pre-sampled set over 5 different runs?

**Ethics Review Area:**

["I don’t know"]

**Limitations:**

In spite of excellent results on easy tasks such as simple text classification, the overall performance is still poor on tough tasks like CoLA. This limitation has to be discussed.

**Strengths And Weaknesses:**

Strength:

The idea of modifying an auto-regressive PLM decision boundary to a discriminative one by generating synthetic data is interesting.

The empirical evaluation shows promising results of SuperGen beating the performance of PLMs used for the few-shot setting.

Weakness:

Although it's encouraging to see that the zero-shot SuperGen outperforms the few-shot setting, I'm not fully convinced by the results. As been widely discussed, the data selection process under a few-shot setting can have a huge impact on evaluation results. Not to mention the impact brought by prompts, decoding strategies, etc. I understand that some of these factors are difficult to control, so I suggest the following baselines for evaluation:

PLMs fine-tuned on the original train set for each task. While I understand that these results will outperform the results of SuperGen, I think it is still relevant and important to know what is the gap in performance between fine-tuning the PLMs and the zero-shot evaluations as shown in the Table.
A knowledge distillation baseline: using the PLM to generate the output label when provided with an unlabeled text input (from some corpus), and then using these pseudo labels to train a discriminative model. This would help shed light on what is the benefit of creating the synthetic data rather than directly transferring knowledge from the PLM to downstream models.

It would have been interesting to see if the generated synthetic data in a zero-shot manner could add to the labeled supervised data for the tasks via data augmentation, and help improve the performance of the fully supervised setting.

---

> ### Author Response · Authors · 2022-08-02
> **Response to Reviewer XfNA**
>
> Thanks for your thoughtful review and feedback! As mentioned in the general response, we have updated the paper with two baselines (fully supervised and knowledge distillation) as you suggest.
> We provide more detailed discussions as follows:
> 1. **Regarding the knowledge distillation baseline.** As shown in Table 4, the zero-shot predictions of CTRL are rather inaccurate, and these results are in line with previous studies: Table 1 in Zhao et al. shows that a 2.7B GPT-3 model achieves 71.4 zero-shot accuracy on SST-2 (even with calibration), while we use a 1.5B CTRL model to achieve 72.5 accuracy on SST-2 with carefully chosen prompts; Table 2 in Wei et al. reports that a 137B LaMDA-PT achieves 35.7/37.0 zero-shot accuracy on MNLI-m/mm, while our CTRL results are 38.0/39.2. These inaccurate predictions fail to serve as good soft labels for training classifiers (even if we directly use the original training set as unlabeled data).
> 2. **Why is using unidirectional PLMs for training data generation a better choice than using them for knowledge distillation?** We hypothesize that there are two major reasons: (1) Better flexibility in prompt formats. Using unidirectional PLMs for generation is not subject to any prompt format constraints. However, when they are used for zero-shot predictions, the prompts have to be designed so that the label word is the last token in the sequence to be predicted, as unidirectional PLMs cannot attend to subsequent tokens. Such constraints may result in the prompt being less natural and dissimilar to the pretraining data distribution, which worsens the prediction quality of the PLMs --- such a hypothesis is supported by the fact that a smaller-size bidirectional PLM (a 356M RoBERTa), which can place the label word to be predicted at any position, has much better zero-shot prompting performance (Table 2 first row) than a 1.6B CTRL. (2) More direct uses of PLMs' language modeling ability. Using unidirectional PLMs for training data generation _directly_ leverages the PLMs' output token probability. Applying PLMs for zero-shot prediction, however, requires an additional step to convert token predictions to label predictions, and such a mapping process usually necessitates manual curation and can hardly be optimal, especially without abundant task-specific data.
> 3. **Use synthetic data to improve fully supervised performance.** Since many GLUE classification tasks come with a large original training set (e.g., MNLI/QQP/QNLI have 393K/364K/108K training samples), we doubt the synthetic data generated by SuperGen may further improve the fully supervised results. However, the few-shot results added in Section 5.5 (suggested by Reviewer kf6s) demonstrate that it is indeed feasible to use synthetic data to improve the classifier performance when the training data are limited.
> 4. **Are these 5 runs performed on 5 different few-shot train/dev sets or on the same pre-sampled set over 5 different runs?** We perform training data generation 5 times under different seeds so that the synthetic data will be different. The average/standard deviation are calculated from the classifier performance trained on these different generated training sets.
> 5. **Poor performance on tough tasks like CoLA.** Generating quality training data for CoLA is indeed challenging because the “linguistically incorrect” label of CoLA requires generating sequences with grammar mistakes, which will be of a different distribution from the one PLMs are trained on. Therefore, it is necessary to rely on some labeled training data – indeed, as shown in Table 5, leveraging a small few-shot training set significantly improves the performance on CoLA. We have added discussions regarding this limitation in Section 6.
>
> Thank you again for your review! Please let us know if you have any further comments.
>
> References:
> Zhao, Tony et al. “Calibrate Before Use: Improving Few-Shot Performance of Language Models.” ICML 2021.
> Wei, Jason et al. “Finetuned Language Models Are Zero-Shot Learners.” ICLR 2022.

---

> > ### Comment · Reviewer_XfNA · 2022-08-09
> > **Lots of missing related works (with similar problem definition, motivation, and even solution)**
> >
> > Dear authors and peer reviewers,
> >
> > Thanks for the comprehensive feedback, which mostly addresses my concerns upon evaluation.
> >
> > However, after a more thorough reading of papers in this field recently, I found lots of missing related works that have not been properly discussed, for instance [1,2,3].
> >
> > [1] Generating Datasets with Pretrained Language Models EMNLP 2021
> > [2] GPT3Mix: Leveraging Large-scale Language Models for Text Augmentation. EMNLP 2021
> > [3] ZeroGen: Efficient Zero-shot Learning via Dataset Generation. 2022.2
> >
> > These existing works share lots of similarities with this submission, e.g., do not require human annotations, generate pseudo data for zero-shot learning, etc.
> > I believe those existing works greatly affect the contribution of this submission, e.g., [3] has an almost identical solution to this submission.
> >
> > Based on this, I would like to change my score from 4 to 3, but I'm happy to hear other reviewers' thoughts about this issue.

---

> > > ### Author Response · Authors · 2022-08-09
> > > **Response to Reviewer XfNA**
> > >
> > > Dear Reviewer XfNA,
> > >
> > > Thanks for your reply! We are glad that our response addressed your concerns, and we are happy to provide explanations regarding the similarity to the related work that you mentioned:
> > >
> > > 1. **Regarding the similar solution to ZeroGen by Ye et al.** We would like to kindly point out that our work was available on arXiv **earlier** than Ye et al. Actually, our paper is even **cited by Ye et al.** Performance-wise, our proposed approach is also better than ZeroGen (e.g., on SST-2, RTE and QNLI). That said, we agree that ZeroGen can be considered as related work, and we have included it in the updated version.
> > > 2. **Regarding the comparison to Yoo et al.** Our data generation method is different from GPT3Mix: GPT3Mix works under the _few-shot_ setting by "mixing" task-specific labeled data to create new samples, and it cannot be used in _zero-shot_ settings where no labeled data are available. On the contrary, our method directly generates training data guided by prompts, without using any labeled samples. In other words, GPT3Mix works by producing samples with similar semantic/syntactic structures to the given few-shot demonstrations, while our SuperGen method directly extracts the pretrained knowledge from PLMs through generation guided by prompts.
> > > Empirically, on SST-2, our _zero-shot_ results even significantly outperform the _few-shot_ results of GPT3Mix which has access to 1% of the training set, ~600 labeled samples. (For CoLA, the results are not directly comparable since GPT3Mix reports accuracy while we report Matthews correlation, the official evaluation metric on CoLA). GPT3Mix also uses a much larger PLM (175B) than ours (1.6B).
> > > 3. **Regarding the comparison to Schick et al.** Schick et al. mainly studied generating similar and dissimilar sentence pairs to train sentence embeddings (for textual
> > > similarity tasks) and their method cannot be directly applied to the wide range of NLU tasks studied in our paper. For example, their generation method obviously cannot be used for question-answering classification tasks (e.g., QNLI) or linguistic acceptability classification tasks (e.g., CoLA).
> > >
> > > Overall, our proposed approach and setting is sufficiently different from Yoo et al. and Schick et al. More specifically, our method effectively utilizes unidirectional PLMs to generate quality training samples _without using any task-specific data_ (this is not the case in Yoo et al.) and _without using gigantic models_ (Yoo et al. rely on very large PLMs like 175B GPT3 to perform well). We also show that our method is generalizable to a wide range of NLU tasks (the method by Schick et al. is specifically for sentence similarity tasks). The paper by Ye et al. was available on arXiv after our paper. We have discussed the three papers as related work in the updated version.
> > >
> > > We would be happy to know your further comments!
> > >
> > > References:
> > > Ye, Jiacheng et al. “ZeroGen: Efficient Zero-shot Learning via Dataset Generation.” ArXiv 2022.
> > > Yoo, Kang Min et al. “GPT3Mix: Leveraging Large-scale Language Models for Text Augmentation.” EMNLP 2021.
> > > Schick, Timo and Hinrich Schütze. “Generating Datasets with Pretrained Language Models.” EMNLP 2021.

---

> > > > ### Comment · Reviewer_XfNA · 2022-08-09
> > > > **comment**
> > > >
> > > > Thanks for the detailed response. I do not have further questions and will raise my score.

---

> > > > > ### Author Response · Authors · 2022-08-09
> > > > > **Thanks for your feedback!**
> > > > >
> > > > > Thanks for your feedback!

---

### Author Response · Authors · 2022-08-02
**General Response**

We sincerely thank the three reviewers for thoughtful comments! We have updated the paper according to reviewers' suggestions, and we summarize the major changes as follows:
1. Add the fully supervised results (Table 2 last row) per Reviewer XfNA.
2. Add the results of using CTRL for zero-shot prompting and for knowledge distillation per Reviewer XfNA (Section 5.4).
3. Add the results of fine-tuning on both few-shot and synthetic data per Reviewer kf6s (Section 5.5).
4. Extend the prompt ablation studies to all tasks per Reviewer V4Sv (Table 3).

---

### Author Response · Authors · 2022-08-08
**Looking Forward to Your Reply!**

Dear Reviewers,

Thank you again for your valuable feedback and comments! Since the discussion period is ending soon, we would greatly appreciate it if you could let us know if you are satisfied with our response. We will be happy to address any remaining concerns.

Sincerely,
Paper11789 Authors

---

### Meta-Review · Area_Chair_sXrs · 2022-08-29

[review text omitted: it was posted to a different submission]

---

### Decision · Program_Chairs · 2022-09-14

Accept